# Influence of Culture Conditions on the Bioreduction of Organic Acids to Alcohols by *Thermoanaerobacter pseudoethanolicus*

**DOI:** 10.3390/microorganisms9010162

**Published:** 2021-01-12

**Authors:** Sean Michael Scully, Aaron E. Brown, Yannick Mueller-Hilger, Andrew B. Ross, Jóhann Örlygsson

**Affiliations:** 1Faculty of Natural Resource Science, University of Akureyri, Borgir v. Nordurslod, 600 Akureyri, Iceland; scully@unak.is (S.M.S.); yannick.mueller-hilger@stud.h-da.de (Y.M.-H.); 2Faculty of Education, University of Akureyri, Solborg v. Nordurslod, 600 Akureyri, Iceland; 3School of Chemical and Process Engineering, University of Leeds, Leeds LS2 9JT, UK; bs11aeb@leeds.ac.uk (A.E.B.); a.b.ross@leeds.ac.uk (A.B.R.)

**Keywords:** biocatalysis, extremophile, thermophile, fusel alcohols, carboxylic acids, volatile fatty acids, bioreduction

## Abstract

*Thermoanaerobacter* species have recently been observed to reduce carboxylic acids to their corresponding alcohols. The present investigation shows that *Thermoanaerobacter pseudoethanolicus* converts C2–C6 short-chain fatty acids (SCFAs) to their corresponding alcohols in the presence of glucose. The conversion yields varied from 21% of 3-methyl-1-butyrate to 57.9% of 1-pentanoate being converted to their corresponding alcohols. Slightly acidic culture conditions (pH 6.5) was optimal for the reduction. By increasing the initial glucose concentration, an increase in the conversion of SCFAs reduced to their corresponding alcohols was observed. Inhibitory experiments on C2–C8 alcohols showed that C4 and higher alcohols are inhibitory to *T. pseudoethanolicus* suggesting that other culture modes may be necessary to improve the amount of fatty acids reduced to the analogous alcohol. The reduction of SCFAs to their corresponding alcohols was further demonstrated using ^13^C-labelled fatty acids and the conversion was followed kinetically. Finally, increased activity of alcohol dehydrogenase (ADH) and aldehyde oxidation activity was observed in cultures of *T. pseudoethanolicus* grown on glucose as compared to glucose supplemented with either 3-methyl-1-butyrate or pentanoate, using both NADH and NADPH as cofactors, although the presence of the latter showed higher ADH and aldehyde oxidoreductase (ALDH) activity.

## 1. Introduction

Beyond biofuels, the sustainable production of chemical building blocks from renewable materials is a major goal of the bioeconomy. In addition to their utility as solvents, alcohols are very useful chemical intermediates in the synthesis of other materials. At present, the vast majority of alcohols are produced via the oxo process from petrochemical starting materials, although large quantities of 1-butanol were produced via fermentation in the early to mid-20th century [1] and the biological production of ethanol from second-generation biomass is an expanding area of intense research. The production higher carbon alcohols from gases (CO/CO_2_) has also been explored for 1- and 2-propanol [2], 2-methyl-1-butanol (iso-butanol), 1-hexanol [3], and 1-octanol [4] using a number of species within the class of Clostridia, although these fermentation routes have not reached industrial viability. Until recently, the production of C3 and higher alcohols has largely been restricted to mesophiles. Instead of producing low molecular weight alcohols, a more feasible approach would be to produce higher-order alcohols as their energy density increases linearly with increasing carbon number. For example, the energy density of ethanol, propanol, and butanol are 26.8, 33.4, and 36.1 MJ kg^−1^, respectively, as compared with 44.4 MJ kg^−1^ for isooctane. Higher-order alcohols, such as butanol, are also more desirable as biofuels as they are less corrosive to existing infrastructure and are less hygroscopic than low molecular weight alcohols [5]. While biobutanol production has a long history using mesophilic *Clostridium* species with the acetone-butanol-ethanol (ABE) fermentation using *Clostridium acetobutylicum* and *Clostridium beijerinckii* dating back to the early 20th century [6,7,8,9], the economic viability of such processes is, at present, limited to the cost of substrates and enzymatic pretreatment of complex biomass. More recently, a wild-type thermophilic Clostridia, *Thermoanaerobacterium* strain M5, [10] has been found to produce butanol from a wide-range of substrates including xylan. These bacteria produce acetate and butyrate in the acetogenic phase but due to the acid formation, the pH drops in the medium and the bacteria convert the acids to acetone and butanol similar to mesophilic Clostridia. As carboxylic acids are ubiquitous in nature, producing alcohols from abundant and inexpensive sources such as waste streams could potentially present a sustainable route to the valorization of these materials.

Thermophilic anaerobic bacteria within the genera of *Thermoanaerobacter*, *Thermoanaerobacterium,* and *Caldanaerobacter* within the class Clostridia have been known for some time for being good ethanol and hydrogen producers as are other Firmicutes such as *Geobacillus thermoglucosidasius* [11,12,13]. The use of these bacteria is especially advantageous when complex biomass is used as a feedstock for fermentation, since they typically have a very broad substrate spectrum degrading various monosaccharides, disaccharides, and oligosaccharides present in these type of biomass to volatile end products [11]. More generally, the use of thermophiles is often advantageous due to higher cultivation temperatures increasing the thermodynamic favorability and decreasing the risk of mesophilic contamination while facilitating the direct removal of volatile end products, many of which have deleterious effects associated with their accumulation [12,14].

*Thermoanaerobacter pseudoethanolicus* strain 39E has been extensively investigated for its biotechnological potential. This strain was isolated in the early 1970s from a thermal feature in Yellowstone National Park (WY, USA) and originally classified as a strain of *Clostridium thermohydrosulfircum* [15] before being transferred to the genus of *Thermoanaerobacter* as a strain of *Thermoanaerobacter ethanolicus* [16]. Further studies based upon sequencing of the 16S rRNA gene and DNA-DNA hybridization established that strain 39E was in fact a novel species within the genus [17]. *T. pseudoethanolicus* has been extensively investigated for its ability to produce ethanol from a wide range of carbohydrates, including xylose, and its thermostable pullulanases [18,19] and glucotransferases [20,21,22]. Other organisms within the genera of *Thermoanaerobacter* have broad biotechnological potential beyond their abilities to produce biofuels. *Thermoanaerobacter* strains in general possess multiple alcohol dehydrogenases (ADHs) which have either specificity for primary (PADHs) or secondary alcohols (SADHs) [23,24,25,26,27]. These ADHs also possess different functions with the PADH serving to oxidize ethanol using NAD^+^ and regenerate reducing potential while the SADH is key for the production of ethanol and uses NADP(H) as a cofactor [25,26,28]. The SADHs of *Thermoanaerobacter pseudoethanolicus* strain 39E^T^ have been investigated for their ability to reduce ketones [24,29] Additionally, an acetaldehyde dehydrogenase (AdhE) catalyzes the NADH-dependent acetyl-CoA hydrolysis and condensation to form acetaldehyde which can undergo subsequent reduction to ethanol [24].

Another approach to the production of butanol and other higher-order alcohols would be the biological reduction of inexpensive carbonyl-containing substrates such as ketones and carboxylic acids. It has been demonstrated that *Thermoanaerobacter* species often possess multiple alcohol dehydrogenases including those that are specific for the oxidation and reduction of PADHs and SADHs. The use of *Thermoanaerobacter* SADHs has been widely exploited for the asymmetric reduction of ketones to chiral alcohols as reviewed elsewhere [30,31,32,33,34,35]. The utility of PADHs beyond their role in ethanol formation [36] has received considerable attention. While PADHs are able to oxidize primary alcohols, presumably to their corresponding aldehydes, the reduction of carboxylic acids to these alcohols has only recently been demonstrated from carboxylic acids using glucose [36,37] or amino acids [38,39] as a source of reducing potential.

While the reduction of carboxylic acids to aldehydes and alcohols can be accomplished using traditional synthetic techniques, these approaches often require stoichiometric quantities of reducing agents [40]. The biological reduction of carboxylic acids to alcohols is preferable to the synthesis of alcohols from petrochemical sources given the abundance and low cost of fatty acids; the reduction of carboxylic acids has been previously studied for a number of aerobic organisms such as *Norcardia* [41], while a number of fungi are known to produce mixtures of ketones and alcohols from fatty acids [42]. There are two enzyme systems for the reduction of carboxylic acids, namely aldehyde:ferredoxin oxidoreductases (AORs) and carboxylic acid reductases (CARs) with the latter using ATP and NADPH to facilitate the reduction [43]. Among Clostridia, the study of carboxylic acid reduction been mostly restricted to the *Morella thermoacetica* which possess both a ferredoxin-dependent, tungsten-containing aldehyde-oxidoreductase [44] and several mesophilic autotrophic Clostridia, such as *Clostridium ragsdalei* [45], *Clostridium ljungdahlii* [46], and *Moorella* (formerly *Clostridium*) *thermoacetica* [47] can also produce primary alcohols from their corresponding carboxylic acid. Among fermentative heterotrophic Clostridia, such as *Thermoanaerobacter* and *Caldanaerobacter* strains, it has been observed that branched-chain amino acid catabolism yields a mixture of branched-chain fatty acids (BCFA) and alcohols (BCOH) for which the reduction of the carboxylic acid to the alcohol has also been observed [48]. In this case, it is not clear if the bacteria first produce the acid before conversion to the alcohol or if the intermediate from the amino acid (α-keto acid) is both oxidized to its corresponding fatty acid and reduced to alcohol. A recent investigation in our laboratory on inhibitory effects of various volatile fatty acids on growth revealed that many *Thermoanaerobacter* species can reduce short-chain fatty acids (SCFAs) to their corresponding alcohols during growth on glucose [37]. It is suggested that *Thermoanaerobacter* species can utilize the reducing power generated by glucose oxidation to reduce the acids to alcohols as has been demonstrated with a number of *Thermoanaerobacter* strains, including *T. pseudoethanolicus* [36,37], and studied with respect to BCOH formation *Thermoanaerobacter* strain AK85 during amino acid catabolism in which the amino acid is the source of reducing potential [38].

As is well known, culture conditions are of great importance for end product formation. For example, manipulation of the liquid–gas phase ratio or adding electron scavenging molecules (such as thiosulfate) to the medium makes it possible to direct the end-products to more reduced (ethanol) or oxidized (acetate and hydrogen) formation during glucose or amino acid fermentation [49]. Thus, manipulating culture conditions in an effort to maximize the formation of alcohols by reduction of carboxylic acids is of interest. The present study examines some of the physiological aspects of *T. pseudoethanolicus* strain 39E^T^‘s ability to reduce C2 to C6 carboxylic acids using ^13^C NMR and enzyme assays. A particular emphasis was placed on the influence of culture conditions such as the influence of pH, liquid-gas phase ratio (L-G ratio), and the ratio of the concentration of glucose on short-chain fatty acids. Also of interest is the rate at which carboxylic acids are reduced to their corresponding alcohols and is investigated kinetically in batch culture as is the inhibitory impact of C2–C6 alcohols accumulation.

## 2. Materials and Methods

### 2.1. General Methods

All materials were obtained from Sigma Aldrich unless otherwise stated. Hungate tubes were acquired from ChemGlass (Vineland, NJ, USA). Nitrogen gas used was of 5.0 quality (<5 ppm O_2_) obtained from AGA gas. Nucleotide cofactors were obtained from Megazyme with the exception of NADPH which was obtained from Sigma-Aldrich (Taufkirchen, Germany).

### 2.2. Culture Media and Organisms

All cultivations were performed in Basal Mineral (BM) medium which consisted of the following on a per liter basis: NaH_2_PO_4_·2H_2_O 3.04 g, Na_2_HPO_4_·2H_2_O 5.43 g, NH_4_Cl 0.3 g, NaCl 0.3 g, CaCl_2_·2H_2_O 0.11 g, MgCl_2_·6H_2_O 0.1 g, yeast extract 2.0 g, resazurin 1 mg, trace element solution 1 mL, vitamin solution (DSM 141) 1 mL, and NaHCO_3_ 0.8 g. Glucose was used as a carbon source at a concentration of 20 mM (3.60 g per L) added after autoclaving from a syringe-filtered (0.22 µm) stock solution stored under nitrogen. The trace element composition was as described earlier [50]. The medium was prepared by adding the buffer to distilled water containing resazurin and then boiled for 10 min followed by cooling under nitrogen flushing (<5 ppm O_2_). The mixture was then transferred to serum bottles (typically 25 mL unless otherwise stated) using the Hungate technique [51,52] and autoclaved at 121 °C for 60 min. All cultivations were performed in serum bottles at pH 7.0 with a liquid–gas phase ratio of 1:1 and incubated at 65 °C without agitation unless otherwise stated. All experiments were conducted in triplicate for 5 days unless otherwise indicated. *T. pseudoethanolicus* strain 39E^T^ (DSM 2355) was purchased from Deutsche Sammlung von Mikroorganismen und Zellkulturen. In all experiments, a 2% *v*/*v* inoculation volume obtained from fresh cultures was used. After cultivation, hydrogen was analyzed by gas chromatography, the cells were then centrifuged (13,000 rpm, 3 min) and the supernatant was stored at −40° prior to further analysis.

### 2.3. Conversion of Fatty Acids to Alcohols in the Presence of Glucose

The strain was cultivated on glucose (20 mM) in the absence and presence of a range of an externally added different fatty acid. The acids evaluated (acetate, 1-propionate, 1-butyrate, 2-methyl-1-propionate, 2-methyl-1-butyrate, 3-methyl-1-butyrate, 1-pentanoate, and 1-hexanoate) were added to give a final concentration of 20 mM. The experiments were performed in anaerobic tubes (18 × 150 mm) with equal volumes of liquid and headspace and incubated for 5 days.

### 2.4. Kinetic Experiments of Carboxylic Acid Product Formation

Kinetic experiments on the fermentation of glucose, with and without added fatty acids (1-butyrate, 3-methyl-1-butyrate) were performed in 125 mL serum bottles with a L-G of 1:1 at 65 °C for an incubation period of 120 h. Periodically, 1 mL samples was removed for the analysis of volatiles end products and 0.2 mL of headspace gas was removed for hydrogen analysis.

### 2.5. Effect of Liquid–Gas Phase Ratio on Carboxylic Acid Reduction

The strain was cultivated in BM containing 20 mM of glucose (a) with addition of 1-propionate (b), 1-butyrate (c), 3-methyl-1-butyrate (d), and 2-methyl-1-butyrate (e). Five different liquid–gas (L-G) ratios were used: 0.05, 0.34, 0.98, 2.08, and 5.40 in 58.9 mL serum bottles; as an example, serum bottle with a nominal volume of 58.9 mL of BM will give a L-G ratio of 1.0.

### 2.6. Effect of Initial pH on Carboxylic Acid Reduction

The strain was cultivated in BM containing 20 mM of glucose (a) with addition of 1-propionate (b), 1-butyrate (c), 3-methyl-1-butyrate (d), and 2-methyl-1-butyrate (e). The strain was cultivated at an initial pH between 5.0 and 8.5 in 0.5 unit increments by adjusted with 6 M NaOH or HCl prior to autoclaving. Cultivations were performed in Hungate tube (18 × 150 mm) with a L-G ratio of 1.0.

### 2.7. Inhibitory Effects of C2–C6 Alcohols on End Product Formation

To investigate the inhibitory effects of fatty acids and alcohols, the strain was cultivated on glucose in the presence of ethanol, 1-propanol, 2-propanol, 1-butanol, 2-methyl-1-propanol, 2-methyl-1-butanol, 1-pentanol, and 1-hexanol. The concentrations used were 0.0, 0.5, 1.0, 2.0, 3.0, 4.0, 5.0, and 7.0% (*v*/*v*). Experiments were performed in Hungate tubes (16 × 150 mm) with L-G ratio of 1:1.

### 2.8. Effect of Glucose and Fatty Acid Ratio on End Product Formation

The strain was cultivated at different concentrations of glucose (0, 10, 20, 30, and 40 mM) in the presence of 1-propionate (20 mM), 1-butyrate (20 mM), or 2-methyl-1-butyrate (20 mM). The experiment was performed in Hungate tubes (16 × 150 mm) with L-G ratio of 1:1.

### 2.9. Nuclear Magnetic Resonance (NMR) Experiment

The strain was cultivated in the presence of 20 mM ^13^C1-labled 3-methyl-1-butyrate and glucose (20 mM) in an 8.7 mL serum bottle (L-G 0.98) for 7 days and analyzed. ^13^C nuclear magnetic resonance (NMR) spectra were obtained using a Bruker AV400 at 298 K after spiking with D_2_O to obtain a signal lock (0.3 mL addition of D_2_O to 1 mL of aqueous sample).

### 2.10. Enzyme Assays

Cells for enzymatic assays were cultivated on 20 mM of short-chain fatty acid (SCFA) + 20 mM glucose in 1 L serum bottles containing 500 mL of medium; cells were harvested after 18 h of cultivation via centrifugation (4700 rpm, 0–4 °C after dithionate addition to afford a final concentration of 5 mg/L) followed by washing three times with degassed 50 mM Tris-HCl (pH 7.5). Cells were suspended in 10 mL of 50 mM Tris-HCl (pH 7.5) to which an equal volume of glass beads (150–212 µm) were added followed by vortexing for 30 s and then by cooling on an ice bath for 2 min; this was repeated three times. Cell debris was removed by centrifugation and the supernatant transferred to nitrogen-flushed serum bottles. ADH and AOR activity assays were immediately performed using the nitro blue tetrazolium (NBT) method of [53] as previously described earlier [38] and the concentration of generated NAD(P)H were determined after 60 min and calculated according to the equation below.
$$ADH\;activity\;\left(\frac{mU}{mL}\right)=\frac{nmol\;NADH}{v\cdot t}=nmol\;NADHx2$$

### 2.11. Analytical Methods

Hydrogen was analyzed with a Perkin Elmer Auto System XL gas chromatograph equipped with a thermo-conductivity (TCD) detector as previously described [54]. Alcohols and volatile fatty acids were measured by gas chromatography using a Perkin-Elmer Clarus 580 gas chromatograph equipped with flame ionization detector (FID) as previously described [54]. Optical density was determined using a Shimadzu UV-1800 at a wavelength of 600 nm (*l* = 1 cm). Protein was quantified using the Lowry assay as described by [55] with minor modifications as described by [38]. Bovine serum albumin (BSA) was used as a standard at a concentration ranging from 0.1 to 1.4 mg/mL.

## 3. Results and Discussion

Recent investigations in our laboratory concerning the ability of *Thermoanaerobacter* species to ferment hydrolysates of carboxylic acid-rich biomass revealed that most species within the genus can reduce carboxylic acids to their corresponding alcohols using glucose as a source of reducing potential. This phenomenon has now been demonstrated with several *Thermoanaerobacter* strains [36] including *T. pseudoethanolicus* [37] and further by one strain isolated in Iceland, *Thermoanaerobacter* strain AK85 [38]. The present investigation examines the ability of *T. pseudoethanolicus* to reduce carboxylic acids to their corresponding alcohols with an emphasis on the influence of culture conditions and the inhibitory impact of alcohols on fermentation as well as the activities of enzymes likely involved in the conversion.

### 3.1. Fatty Acid Conversion to Alcohols

To investigate the range of short-chain fatty acids that could be reduced by *T. pseudoethanolicus*, the strain was cultivated on 20 mM glucose in the absence or presence of 20 mM of various C1 to C8 SCFAs, including several branched-chain fatty acids, as shown in Figure 1. Formate, heptanoate, and octanoate were not reduced while the C3–C6 SCFA were reduced with conversions between 21.0–59.7% which is similar to the values reported earlier for this strain [37]. While acetate reduction to ethanol is difficult to detect due to *T. pseudoethanolicus* being highly ethanologenic, previous work does indeed confirm that acetate can be reduced to ethanol [38]. The conversion yields of the C3–C6 SCFAs examined varied; the addition of 1-butyrate resulted in the highest conversion to its corresponding alcohol, 1-butanol (57.9%; 11.58 mM) but only 21% (4.2 mM) of the 3-methyl-1-butyrate was converted to 3-methyl-1-butanol (Figure 1).

The catabolism of glucose without addition of fatty acids resulted in the production of ethanol as the main end product (35.1 ± 1.3 mM or 87.8% of the theoretical yield) with other end products being acetate (6.9 ± 0.1 mM) and hydrogen (3.7 ± 0.5 mmol/L). The end product formation from glucose, in the presence of fatty acids show a decrease in ethanol formation (35.1 mM in the glucose control as compared to 16.5 mM when 1-propionate is included) and an increase in acetate (9.4 mM on 3-methyl-1-butyrate to 22.3 mM when cultivated on 3-methyl-1-propionate) along with the conversion of the exogenously added carboxylic acid to the corresponding alcohol (Figure 1) when exogenous fatty acids are provided. The decrease in ethanol formation suggests that reducing equivalences from glucose oxidation is being redirected from ethanol towards the reduction of the SCFA while the remaining carbon from glucose metabolism is being shunted towards acetate allowing additional ATP formation. This is evidenced by higher optical densities when SCFAs were provided (Figure 1) with the exception of formate which was not reduced to methanol and resulted in a mild inhibition as evidenced by both lower end production formation and a drop in optical density (OD). This is similar to the result reported by Hitschler et al. [38] in which the addition to 2-methyl-1-propionate to glucose-containing fermentations resulted in an increase in OD from 0.58 to 0.80.

Generally, the range of SCFAs reduced by *T. pseudoethanolicus* is quite limited compared to other organisms, such as *Mycobacterium marinum* which can convert C6-C18 fatty acids to alcohols [56]. There are relatively few reports of reduction of carboxylic acids by Clostridia and those that have been reported are mostly autotrophic (Table 1). The whole cell system of *Moorella thermoacetica* can also reduce a wide range of carboxylic acids including C3-C5 fatty acids as well as much more sterically bulky aromatic fatty acid derivatives [47]. The results obtained in this study are very similar to those recently reported by [36] for *Thermoanaerobacter* strains using a whole-cell system (Table 1). These authors also reported a titer of approximately 15 mM of 2-methyl-1-propanol (30% conversion) produced from 50 mM of the corresponding fatty acid for *T. pseudoethanolicus* using a 1:2 glucose:SCFA ratio as compared to 9.9 mM (49.7% conversion) from a 1:1 glucose:SCFA ratio. This suggests that the ratio of reducing equivalence can be used to manipulate the conversion of SCFA to its corresponding alcohol. This is further investigated and discussed in later sections.

### 3.2. Kinetic Experiments

To better understand the conversion of SCFAs to alcohols by *T. pseudoethanolicus*, fermentations were monitored kinetically on 20 mM glucose in the absence and presence of 1-butyrate and 3-methyl-1-butyrate (both 20 mM) as model compounds. Kinetic growth in controls (yeast extract only) are shown in Appendix A. During growth on glucose without acid addition, the strain reached a maximum OD after 18 h, producing 35.0 mM of ethanol, 7.7 mM of acetate, and 2.6 mmol/L of hydrogen (Figure 2A) with the maximum ethanol productivity of 2.0 mmol L^−1^ h^−1^). The degradation of glucose in the presence of 1-butyrate (20 mM) reached a maximum OD within 18 h and, once again, the dominant end products were ethanol, acetate, and hydrogen. As before, 1-butyrate was reduced to 1-butanol; the amount of the acid consumed was 13.6 mM which resulted in production of 13.0 mM of the alcohol or a conversion of 65.0% (Figure 2B). The concentration of ethanol (26.2 mM) was lower compared with the fermentation of glucose alone but the concentration of acetate was higher (15.9 mM). The ethanol productivity was 1.41 mmol L^−1^ h^−1^, considerable lower (30%) as compared with fermentation of glucose only, while the observed maximum productivity for the conversion of butyrate to 1-butanol was 0.94 mmol L^−1^ h^−1^. The decreased ethanol productivity can likely be due to the flow of electrons being diverted to 1-butyrate reduction.

Similarly, the degradation of glucose in the presence of 3-methyl-1-butyrate resulted in the same end products as before (ethanol, acetate, and hydrogen) in similar concentrations as during the growth on glucose in the presence of 1-butyrate. 3-Methyl-1-butyrate was converted to 3-methyl-1-butanol although to a lesser extent as compared with conversion of 1-butyrate to 1-butanol (Figure 2C). The concentration of 3-methyl-1-butyrate decreased from 20 mM to 13.0 mM and 6.2 mM of 3-methyl-1-butanol were produced. As with glucose, the maximum OD was also reached within 18 h suggesting that SCFA conversion occurs rapidly when reducing equivalence are available. The maximum ethanol and 3-methyl-1-butanol productivity was 1.90 and 0.61 mmol/L/h, respectively.

### 3.3. Influence of Initial pH and Partial Pressure of Hydrogen

The end product formation profile of a cultivation of the strain can be easily manipulated by alternating the culture parameters. To investigate the effect of pH and L–G ratio on the conversion of carboxylates to their corresponding alcohols, *T. pseudoethanolicus* was cultivated on glucose containing one of four model carboxylic acids, 1-propionate, 1-butyrate, 2-methyl-1-butyrate, and 3-methyl-1-butyrate. The effect of initial pH did not have a dramatic effect on the resultant alcohol/carboxylic acid ratio although a pH value of around 6.5 appears to be optimal for the alcohol formation (Figure 3A). The influence of initial pH was most pronounced in the case of 1-butyrate reduction; the percentages of 1-butyrate reduced to 1-butanol ranged from 28.1% (at pH 5.0) to 50.0% at pH 6.5 while only 44.6% of the 1-butyrate is converted at pH 7.0. Similarly, the other carboxylic acids showed the highest conversion to their corresponding alcohols at an initial pH of 6.5 with the notable exception of the reduction of 2-methyl-1-butyrate which showed the highest conversion was at an initial pH of 6.0 (24.1% conversion).

It is well known that the partial pressure of hydrogen (*p*H_2_) strongly effects the ratio of oxidized and reduced end product formation with some strains of *Thermoanaerobacter* [60,61,62,63,64]. Thus, at high *p*H_2_, the tendency is to produce more reduced end products like ethanol and lactate but less acetate and hydrogen. During growth on glucose alone, the strain produced a mixture of ethanol and acetate together with hydrogen (Appendix A). At the lowest L-G ratio used, the ratio of ethanol and acetate was 1.76. During other conditions (growth on glucose with supplementary fatty acids) more ethanol and less acetate were produced with increasing L-G ratios (ethanol-acetate ratio varied from 3.51 (0.34 L-G ratio) to 8.95 (at 5.26 L-G ratio) (Appendix A).

During growth of the strain on glucose in the presence of 1-propionate (20 mM) the strain produced 13.5 mM of ethanol and 14.7 mM at the lowest L-G ratio applied (Figure 3B). The concentration of ethanol in the presence of 1-propionate was however much less as compared with glucose alone but visa verse for acetate (Appendix A). This indicates that the electrons are directed away from ethanol formation but towards the reduction of 1-propionate to 1-propanol. Ethanol concentrations were stable at higher L-G ratios (between 9.1 to 11.2 mM or about a one third compared to the glucose control) whereas acetate production decreased with increasing L-G ratios (from 14.7 to 3.6 mM) clearly suggesting that reducing equivalence are being redirected to carboxylic acid reduction (Appendix A). The amount of 1-propionate that was converted to 1-propanol was similar under different L-G ratios and approximately 50% of the acid was converted to the alcohol.

During growth on glucose in the presence of 1-butyrate the strain produced more ethanol as compared with the addition of 1-propionate; between 20.6 mM to 24.7 mM of ethanol, which is approximately two-thirds of ethanol concentrations in the glucose control, and between 14.0 and 19.4 mM of acetate were produced (Appendix A). 1-Butyrate was converted to similar amounts of 1-butanol at all L-G ratio conditions (1-butanol concentrations ranged from 9.9 to 11.2 mM; Figure 3B). Again, acetate concentrations were highest at low L-G ratios. The differences in the amount of 1-propanol and 1-butanol produced may suggest differences in the specificity of the alcohol dehydrogenases involved as has been previously reported for other clostridial solvent producers [23,64].

During the degradation of glucose in the presence of 3-methyl-1-butyrate and 2-methyl-1-butyrate, similar trends in ethanol and acetate were observed as before although ethanol concentrations were higher as compared with addition of 1-propionate and 1-butyrate (Appendix A). However, less of the BCFAs were converted to their corresponding alcohol. Maximum concentrations of 3-methyl-1-butanol and 2-methyl-1-butanol were 5.9 mM and 8.1 mM, respectively (Figure 3B).

Overall, the impact of L–G ratio on the amount of carboxylic acid converted to the corresponding fatty acid appears to be limited; the small quantities of hydrogen produced by *T. pseudoethanolicus* may limit the impact of hydrogen accumulation. Exogenously added hydrogen may be an option to supplement the reducing potential generated by glucose oxidation and drive further carboxylic acid reduction.

### 3.4. Effect of the Ratio of Glucose and Fatty Acid Concentration on Alcohol Formation

In an effort to increase the amount of carboxylic acid converted to its corresponding alcohol, the use of additional reducing potential by increasing the initial concentration of glucose was investigated on three model carboxylic acids: 1-propionate, 2-methyl-1-propionate, and 1-butyrate. Figure 4A–C) shows the increase in the conversion of SCFA to alcohol from 0 to 40 mM glucose which corresponds to a molar ratio of 0.5 to 2 of glucose to SCFA.

Without addition of any glucose, 3.41 mM of 1-propanol were produced from 1-propionate, presumable by using electrons produced from oxidation of substrates present in the yeast extract (Figure 4A). Increased glucose concentrations increased the amount of the alcohol from the acid, reaching a maximum at 40 mM glucose with 74.1% conversion. Clearly, there is a substrate inhibition that occurs between 20 and 30 mM of glucose as reflected in the decrease in the amount of ethanol produced per mole of glucose degraded with similar results being obtained with increasing initial glucose concentrations without the addition of exogenously added SCFA. It is well known that many thermophiles are strongly inhibited at moderate (20–30 mM) initial substrate concentrations [11]. It is unlikely that this decrease of glucose degradation is caused by the addition of the alcohol since 20 mM of 1-propanol is only 0.12% and as stated below the strain tolerates up to 3% 1-propanol concentration. Similar results were also obtained from cultivation of the strain in the presence of 1-butanol and 2-methyl-1-propanol; maximum conversion were observed at the highest concentration of glucose applied, or 50.7% and 53.4%, respectively (Figure 4B,C). Also, similar trends in less glucose degradation as with the culture amended with 1-propanol were observed.

To achieve higher conversions, another culture mode such as fed-batch or continuous culture maybe more appropriate. Furthermore, the use of a biphasic system to remove the SCOHs produced from the fermentation broth may increase titers. This approach has proven successful with butanologenic Clostridia using ionic liquids [65], in situ vacuum distillation [66], and fed-batch supplemented with a non-ionic surfactant such as Tween 80 [67,68].

### 3.5. Inhibitory Impact of the Alcohols on End Product Formation from Glucose

Understanding the impact of the conversion of SCFAs to their corresponding alcohols is necessary to determine at which alcohol concentration become toxic to cells. It is well known that alcohols in general inhibit bacteria by disrupting the cell membrane when in high enough concentrations [69,70]. The severity of inhibition by alcohols is more apparent with higher order alcohols such as butanol as compared with ethanol. For example, most *T. pseudoethanolicus* can tolerate ethanol up to 2.5% but only 1% *v*/*v* 1-butanol [69] although some strains of *T. ethanolicus* can tolerate up to 10% *v*/*v* of ethanol [71]. In order to investigate the inhibitory effects of aliphatic alcohols, *T. pseudoethanolicus* was cultivated on glucose (20 mM) in the presence of exogenously added alcohols (C2 to C6 alcohols) for 5 days as shown in Figure 5A–H. It should be noted that OD data for the higher-order alcohols is not given due to their poor solubility and clouding in aqueous solution.

In case of ethanol, 1-propanol and 2-propanol, ethanol is not shown in Figure 5A–C, either because it was exogenously added or because both propanol types co-elute with ethanol on the GC. The strain tolerated ethanol up to 4% concentration, and in fact hydrogen and acetate production was considerably higher at concentrations between 0.5–3% *v*/*v* as compared with control (no added ethanol) (Figure 5A). The ethanol tolerance here is higher than the result reported previously by Burdette et al. [24] who reported decreased growth rates at least than 1.5% *v*/*v*. At 4% *v*/*v* and higher ethanol concentration end product formation was gradually lower until at 7% when a complete inhibition occurred. Similarly, higher hydrogen and acetate concentrations were observed on 1-propanol when added between 0.5–2.0% *v*/*v* concentration although a complete inhibition of glucose for both alcohols were observed at 3% and higher concentrations (Figure 5B). The addition of 2-propanol resulted in a complete inhibition at 4% *v*/*v* and somewhat higher hydrogen and acetate production was observed at 2.0–3.0% *v*/*v* as compared with cultivation on glucose without acid addition (Figure 5C). It is known that 1-butanol strongly inhibits most bacteria at concentrations around 1% *v*/*v* [72,73], and this was reflected in the results with *T. pseudoethanolicus* in present study. The strain produced higher amounts of hydrogen and acetate at 0.5% *v*/*v* 1-butanol concentrations but was inactive at higher concentrations (Figure 5D). The remaining alcohols tested (2-methyl-1-propanol, 1-pentanol, 2-methyl-1-butanol, 1-hexanol) all inhibited the strain at concentration of 0.5% (*v*/*v*), except for 2-methyl-1-butanol (1.0% *v*/*v*) (Figure 5E–H).

### 3.6. ^13^C NMR Studies

Isotopically labelled studies of the conversion of ^13^C1 acetate, ^13^C1 1-propionate, and ^13^C1 1-butyrate have been previously reported for *T. pseudoethanolicus* (Scully et al. 2019). Here, we demonstrate that ^13^C1 3-methyl-1-butyrate is also converted to its corresponding alcohol in the presence of glucose (Figure 6). The appearance of a new peak (δ 60.3 ppm) can be attributed to the formation of 3-methyl-1-butanol. This is similar to the production of 3-methyl-1-butanol from ^13^C2-labeled leucine recently reported (Scully and Orlygsson, 2019a; Scully and Orlygsson, 2019b). Using ^13^C1 butyrate as a model, the fermentation kinetics were monitored over 96 h of fermentation as shown in Figure 7. ^13^C1 1-butyrate (183.5 ppm) is rapidly converted to the ^13^C1 1-butanol (60.3 ppm) achieving maximum conversion between 16 and 24 h (Figure 7) which is similar to the kinetic experiment performed earlier (Figure 2B). Interestingly, the ^13^C1-butanol peak decreases slightly after 30 h, which corresponds with a similar increase in the ^13^C-1-butyrate peak, suggesting that the fermentation mixture is not yet at equilibrium.

### 3.7. Enzymatic Assays 

*T. pseudoethanolicus* was cultivated on glucose (20 mM) with and without the addition of C5 fatty acids (3-methyl-1-butyrate and pentanoic acid) for 24 h at which time the activities of alcohol dehydrogenase (ADH) and aldehyde oxidoreductase (ALDH) were determined with both NAD^+^ and NADP^+^ as cofactors as summarized in Figure 8A–C. The ALDH activity is likely the result of an AOR as previously suggested by [36] and is common with other carboxylic acid-reducing Clostridia such as “*Clostridium ragsdalei”* [46]. Generally, the ADH and ALDH activities were higher in glucose-grown cultures containing exogenously added fatty acids. On glucose only, the activity of ADH for NAD ranged from virtually zero (with 3-methyl-1-butanol) to 35.9 mU/mg protein (with 1-pentanol). Testing from NADP resulted in range of enzyme activity from 18.3 mU (1-octanol) to 90.8 mU/mg protein. In the presence of 3-methyl-butyrate, there was a considerable increase in activity of both NAD and NADP for ALD. The highest NAD activity was actually observed on ethanol (126.3 mU/mg) but on 1-heptanol (128.2 mU/mg) for the NADP activity. The addition of 1-pentanoate resulted in a slightly lower increase in both NAD and NADP for ALDH activity. Values for NAD activity ranged from 45.2 mU/mg (with 2-propanol) to 77.4 mU/mg (with pentanol). Similarly, values for NADP ranged from 62.3 to 167.3 mU/mg with 2-propanol and 1-heptanol, respectively. Interestingly, activity towards C7 and C8 substrates was detected even though earlier experiments with exogenously added heptanoic and octanoic acid did not produce the corresponding alcohols.

Many *Thermoanaerobacter* strains, including *T. pseudoethanolicus*, have multiple ADHs, including both primary and secondary-specific ADHs which differ in their cofactor specificity [23,26,35]; both primary specific and secondary specific ADHs are active under the examined conditions. This may suggest that higher ADH expression can be achieved by adding SCFAs to the fermentation medium. Interestingly, lower ALDH activities on C2–C4 aldehydes are observed using NAD^+^ as a cofactor in the C5-alcohol grown cells as compared with the glucose controls; for example, ALDH activities towards propionaldehyde are 42.6, 67.9, and 71.4 mU/mg of protein for cells grown on glucose, glucose and 3-methyl-1-butyrate, and glucose and 1-pentanoate, respectively. Interestingly, the ADHs present seem to be able to oxidize 1-heptanol and 1-octanol although the corresponding AOR activity was not examined. It should be noted that C7 and C8 alcohols are poorly soluble in aqueous systems so non-aqueous or biphasic systems may be necessary to achieve reduction of the corresponding C7 and C8 carboxylic acids.

While the conversions of carboxylic acids to their corresponding alcohols is lower than some of those reported by mesophilic acetogens, the ability of *T. pseudoethanolicus* to perform these reductions quickly using reducing potential from carbohydrates presents a substantial advantage over autotrophic systems for which the solubility of gases becomes limiting. Forthcoming work will specifically examine the possibility of using a fed-batch approach coupled with in situ alcohol extraction as a means of improving alcohol yields from carboxylic acid reduction. While a number of challenges remain, such as the toxicity of the higher-order alcohols, the use of other sources of reducing potential is interesting. An area of deserving further scrutiny is the nature of the metabolic intermediate during carboxylic acid reduction as well as the path that the reducing potential takes during the course of carboxylic acid reduction. Future efforts will focus on techniques to mitigate the toxicity of alcohol accumulation and directing more of the reducing potential towards carboxylic acid reduction perhaps through the elimination of competing pathways, such as those involved in hydrogen and lactate production.

## 4. Conclusions

*T. pseudoethanolicus* is a promising strain for the reduction of carboxylic acids as it rapidly converts C2–C6 short-chain fatty acids to their corresponding alcohols in the presence of glucose under slightly acidic conditions. Other culture conditions such as liquid–gas phase ratio have little impact on the conversion of the short-chain fatty acids to alcohols although providing additional reducing potential (glucose) increases the amount converted. C4 and higher alcohols are particularly inhibitory to *T. pseudoethanolicus* suggesting that other culture modes may be necessary to improve the amount of fatty acids converted. The use of *T. pseudoethanolicus* for the reduction of carboxylic acids present a potentially useful route to the production of alcohols from inexpensive carboxylic acids.

## Figures and Tables

**Figure 1 microorganisms-09-00162-f001:**
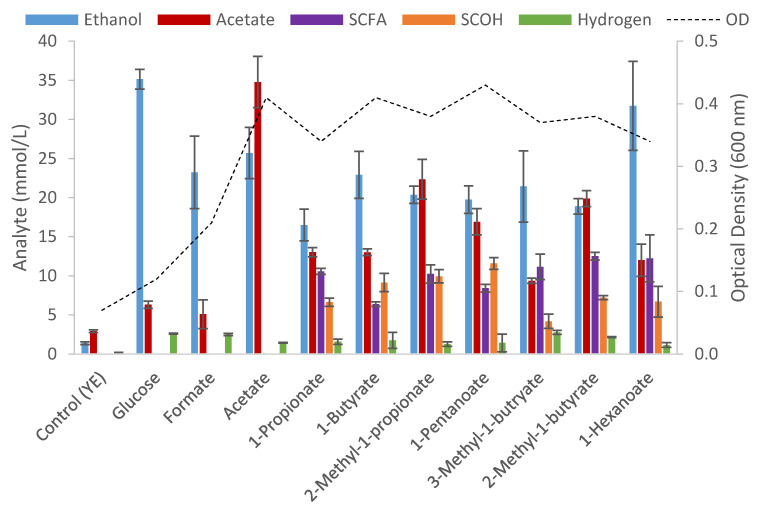
End product formation after 5 days cultivation from cultures of *T. pseudoethanolicus* containing glucose (20 mM) and of exogenously added carboxylic acid (20 mM) and its conversion to its corresponding short-chain alcohol (SCOH). Values represent the average of triplicate fermentations with standard deviation presented as error bars.

**Figure 2 microorganisms-09-00162-f002:**
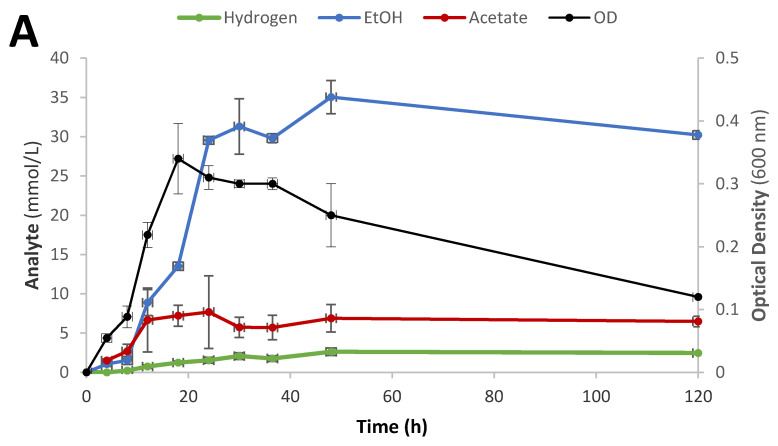
Time-course studies of fermentation of 20 mM glucose (**A**), 20 mM 1-butyrate + 20 mM glucose (**B**), and 20 mM 3-methyl-1-butyrate + 20 mM glucose (**C**) by *T. pseudoethanolicus*. Values represent the average of triplicate fermentations with standard deviation presented as error bars.

**Figure 3 microorganisms-09-00162-f003:**
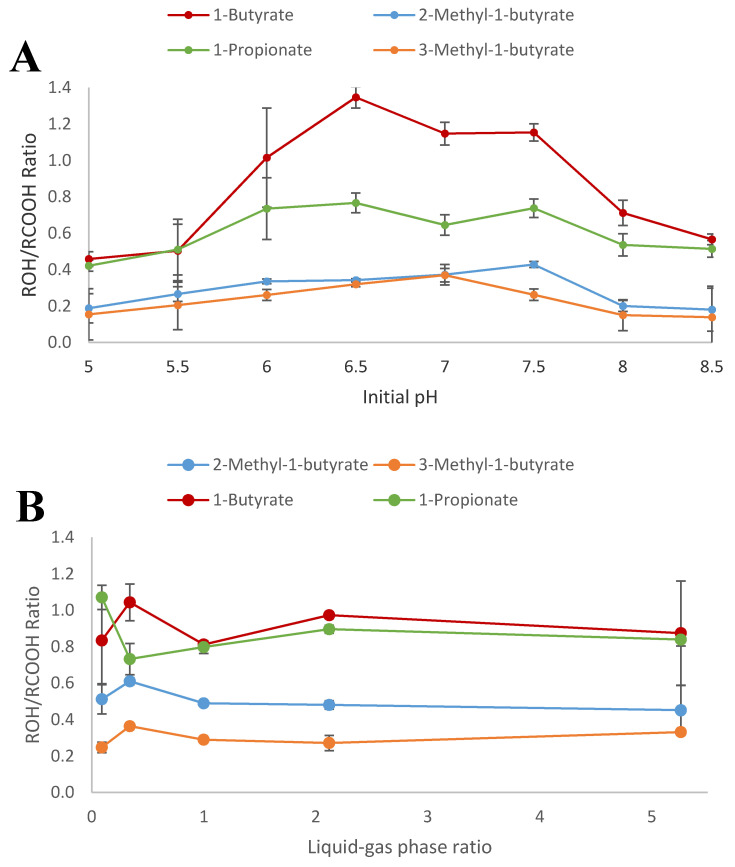
Influence of initial pH (**A**) and liquid–gas phase ratio (**B**) on cultures of *T. pseudoethanolicus* grown on glucose (20 mM) supplemented a carboxylic acid (20 mM). Values represent the average of triplicate measurements ± standard deviation.

**Figure 4 microorganisms-09-00162-f004:**
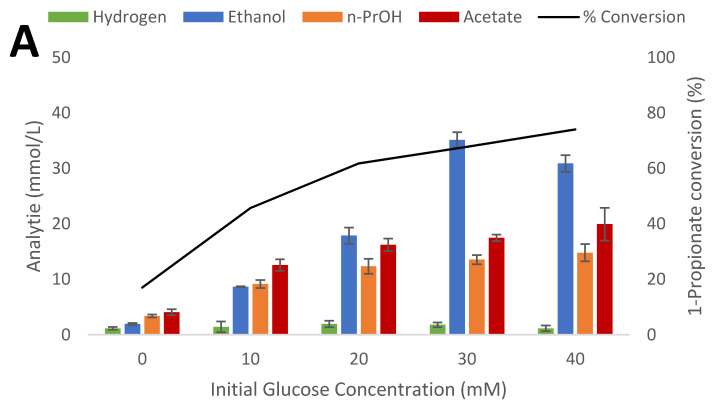
Impact of glucose concentration on end product formation by *T. pseudoethanolicus* in the presence of (**A**) 1-propionate (**B**) 1-butyrate (**C**) 2-methyl-1-propionate bioconversion. Additionally, the percent of glucose consumed is shown (%C). Standard deviation is presented as error bars.

**Figure 5 microorganisms-09-00162-f005:**
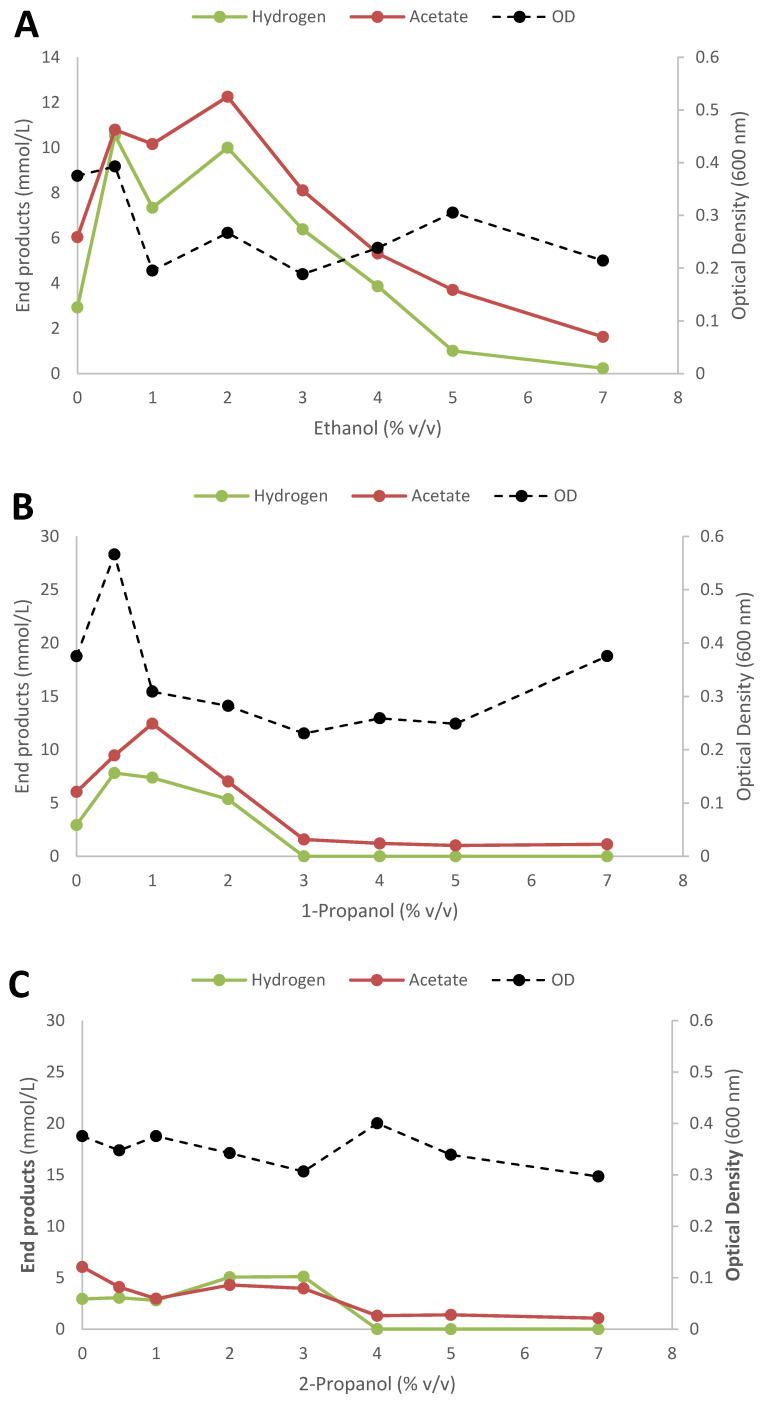
Impact of alcohol addition on end product formation of glucose fermentation by T. pseudoethanolicus using (**A**) ethanol, (**B**) 1-propanol, (**C**) 2-propanol, (**D**) 1-butanol, (**E**) 2-methyl-1-propanol, (**F**) 2-methyl-1-butanol, (**G**) 1-pentanol, (**H**) 1-hexanol. Values represent the average of triplicate measurements ± standard deviation.

**Figure 6 microorganisms-09-00162-f006:**
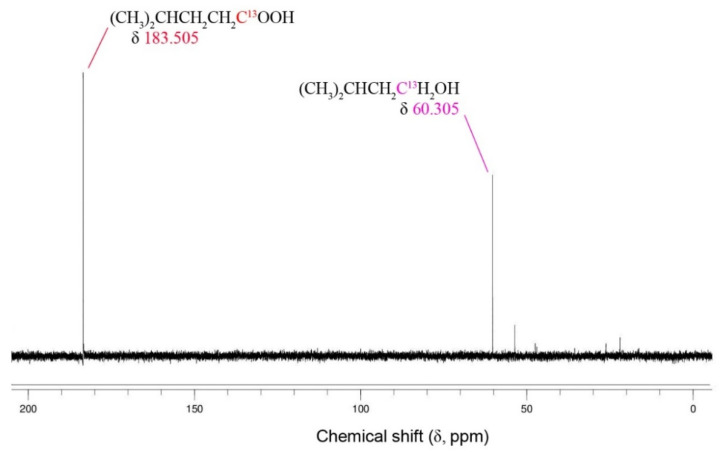
Conversion of ^13^C1 3-methyl-1-butyrate to 3-methyl-1-butanol by *T. pseudoethanolicus* using glucose as the source of reducing potential.

**Figure 7 microorganisms-09-00162-f007:**
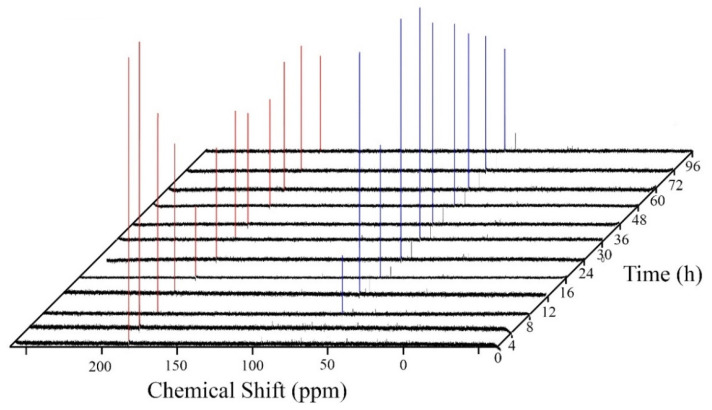
Time course of ^13^C1 1-butyrate (red) conversation 1-butanol (blue) by *T. pseudoethanolicus* using glucose as the source of reducing potential.

**Figure 8 microorganisms-09-00162-f008:**
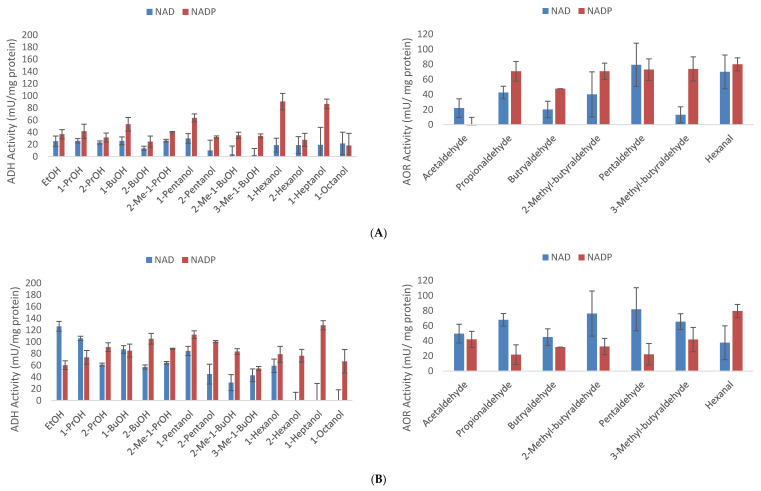
Alcohol and aldehyde dehydrogenase activities of *T. pseudoethanolicus* cells grown on 20 mM glucose (**A**), 20 mM 3-methyl-1-butyrate + 20 mM glucose (**B**), 20 mM 1-pentanoate + 20 mM glucose (**C**). Values represent the average of triplicates with standard deviation presented as error bars.

**Table 1 microorganisms-09-00162-t001:** Conversion of exogenously added carboxylic acids to their corresponding alcohols by selected Clostridia.

Organism(s)	Substrate	Alcohol Titer (mM)	%c	References
*Alkalibaculum bacchi* strain CP15	Syngas + 1-propionate	6.7 (1-PrOH)	36.8	[57]
*Alkalibaculum bacchi* strain CP15+ *Cl. propinicum*	Syngas + 1-propionate	16.6 (1-PrOH)	83.4	[57]
*Alkalibaculum bacchi* strain CP15	Syngas + 1-butyrate	6.7 (1-BuOH	38.6	[57]
*Alkalibaculum bacchi* strain CP15+ *Cl. propinicum*	Syngas + 1-butyrate	11.0 (1-BuOH)	74.7	[57]
*Alkalibaculum bacchi* strain CP15	Syngas + 1-hexanoate	7.9 (1-Hexanol)	63.6	[57]
*Alkalibaculum bacchi* strain CP15+ *Cl. propinicum*	Syngas + 1-hexanoate	9.8 (1-Hexanol)	90.7	[57]
*“Clostridium ragsdalei”*	CO + propionate (15 mM)	7.5 mM (1-PrOH)	30%	[46]
*“Clostridium ragsdalei”*	CO + propionate (30 mM	29 mM	97%	[45]
*C. ljungdahlii* ERI-2	CO + propionate (15 mM)	10.4 mM (1-PrOH)	69.4	[46]
*Clostridium butylicum*	Glucose (36 mmol)	22 mM (1-BuOH	n/a	[58]
*Clostridium butylicum*	Glucose (36 mmol) + 1-butyrate (15 mmol)	26 mM (1-BuOH)	26.7 ^a^	[58]
*T. pseudoethanolicus*	Glu + 1-propionate	6.62 (1-PrOH)	33.0	[37]
*T. pseudoethanolicus*	Glu + 1-butyrate	9.14 (1-BuOH)	55.6	[37]
*T. pseudoethanolicus*	Glu + 1-hexanoate	6.69 (1-Hexanol)	33.5	[37]
*Thermoanaerobacter* strain AK152	Glu + 1-propionate (pH 6.7)	11.5 (1-PrOH)	57.3	[59]
*Thermoanaerobacter* strain X514	Glu + 1-propionate	25 mM (1-PrOH	50	[36]
*Thermoanaerobacter* strain X514	Glu + 1-butryate	<2 mM (1-BuOH)	<4	[36]
*Thermoanaerobacter* strain X514	Glu + 1-hexanoate	8 mM	16	[36]
*T. brockii* subsp. *finnii*	Glu + 1-propionate	21 mM (1-PrOH)	42	[36]
*T. brockii* subsp. *finnii*	Glu + 1-butryate	<5 mM	<10	[36]
*T. brockii* subsp. *finnii*	Glu + 1-hexanoate	<5 mM	<10	[36]

^a^ Amount produced from glucose alone subtracted. n/a-Not applicable.

## Data Availability

Data available in a publicly accessible repository.

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
