# Peer review of "Influence of Culture Conditions on the Bioreduction of Organic Acids to Alcohols by Thermoanaerobacter pseudoethanolicus"

_microorganisms, 2021, doi:10.3390/microorganisms9010162_

Round 1

Reviewer 1 Report

Understanding the reduction of carboxylic acids to their alcohols in thermophile is of great interest in metabolic engineering. The manuscript entitled “Influence of Culture Conditions on the Bioreduction of Organic Acids to Alcohols by Thermoanaerobacter pseudoethanolicus” characterized the physiological culture conditions of T. pseudoethanolicus to reduce C2 to C6 carboxylic acids. The results from this manuscript demonstrate the useful insights of the utilization of T. pseudoethanolicus in the production of alcohols. In most cases, the experiments were well designed. However, the manuscript has some weaknesses in the data presentation. Additional edits of this manuscript in terms of language and structure would make it more presentable for publication. I have a few suggestions below to improve the work.

  1. All the figures lack statistical analyses. Proper significance tests should be added, with descriptions in the methods and figure legends.
  2. In the Figure 1, define SCFA and SCOH in the figure legends. Does SCOH represent the corresponding alcohol of the fatty acid or a mixture of several alcohols? Maybe also clarify that in the legends.
  3. Figure 2 lacks a label for the x axis.
  4. Figure 5 lacks error bars.
  5. The chemical shift profile in Figure 7 is hard to read. Maybe redraw the figure with thicker lines.

Author Response

  1. All the figures lack statistical analyses. Proper significance tests should be added, with descriptions in the methods and figure legends.

During the writing process, the authors vigorously discussed providing a more detailed statistical analysis but ultimately decided against it. The authors feel that detailed stats in this instance do not help with analyzing the end product data and may actually obfuscate the presentation of the data.

  1. In the Figure 1, define SCFA and SCOH in the figure legends. Does SCOH represent the corresponding alcohol of the fatty acid or a mixture of several alcohols? Maybe also clarify that in the legends.

Agreed. The SCOH represents only the corresponding alcohol of the carboxylic acid added. This has now been clarified in text and in the figure caption

  1. Figure 2 lacks a label for the x axis.

This has been fixed. The x- and y-axis was also missing major tick marks and the time interval has been tightened up to prevent the awkward spaces after 120 h

  1. Figure 5 lacks error bars.

Error bars have been added

  1. The chemical shift profile in Figure 7 is hard to read. Maybe redraw the figure with thicker lines.

Agree; Figure 7 is very hard to read upon re-reading. The figure has been re-drawn with a thicker line weight (3 pt )and further clarified by color coding the 13C1 peak for butyrate and butanol red and blue, respectively.

Reviewer 2 Report

The present study devote to some of the physiological aspects of T. pseudoethanolicus ability to reduce C2 to C6 carboxylic acids.

My comments:

First of all, the type strain, 39ET (=DSM 2355=ATCC 33223), was isolated from the Octopus Spring algal–bacterial mat inYellowstone National Park, WY, USA and characterized by Zeikus et al.(1980) as a strain of Clostridium thermohydrosulfuricum (Hollaus & Sleytr, 1972) and was later described as a Thermoanaerobacter ethanolicus strain, 39E (Lee et al.,1993). In 2007 it was renamed as Thermoanaerobacter pseudoethanolicus. Thus, everywhere you need to write T. pseudoethanolicus strain 39ET.

The introduction is written very broadly, but it remains unclear why it is necessary to obtain alcohols using thermophilic bacteria, if there are industrial methods for producing alcohols using yeast. In my opinion, it is necessary to analyze the information about the strain 39ET that has been known since the end of the last century, when it was widely studied as a potential biotechnological agent for ethanol production.

References are given very carelessly in the text. So, in the sentence (Lines 53-54) “Thermophilic anaerobic bacteria within the genera of Thermoanaerobacter, Thermoanaerobacterium, and Caldanaerobacter have been known for some time for being good ethanol and hydrogen producers” [11–13] no mention of Geobacillus thermoglucosidasius from ref 13.

  1. Cripps, R.E.; Eley, K.; Leak, D.J.; Rudd, B.; Taylor, M.; Todd, M.; Boakes, S.; Martin, S.; Atkinson, T. 84 Metabolic engineering of Geobacillus thermoglucosidasius for high yield ethanol production. Metab. Eng. 85 2009, 11, 398–408, doi:10.1016/j.ymben.2009.08.005.

Line 117 Growth medium is glucose and peptone free and is unlikely to support the growth of such saccharolytic bacterium as T. pseudoethanolicus strain 39ET.

Line 125-126 T. pseudoethanolicus strain 39ET = DSM 2355T type strain of species.

Line 146 Effect of initial pH on carboxylic acid reduction - how the initial pH was set? How did the pH change at the end of cultivation?

Line 131 What medium was used?

Line 136 What medium was used for kinetic experiments?

line 142 and 147. What is BM?

Why did the cultivation continue for 5 days if the doubling time of the cell population of strain 39E is 75 min at 65oC? On all curves after 40 hours the optical density dropped, which indicates sporulation or cell lysis.

Cultivation with yeast extract as a carbon source cannot be used as control because yeast extract does not support the growth of the strain.

Line 207 It is known that lactate is another product of glucose fermentation by strain 39ET. Was lactate formed in your experiments?

As known strain 39ET ferments not only glucose, but carbohydrates include xylose, cellobiose, starch, maltose and sucrose. What is known about the formation of alcohols from the corresponding organic acids in the presence of other carbohydrates?

From page 15, there is no line numbering and references are not drawn up according to the rules.

In the references, about 23% of references related to the studies of the authors; I don’t think this is acceptable.

In my opinion, the text should be rewritten vastly. Only in this case can it be accepted.

Author Response

The introduction is written very broadly, but it remains unclear why it is necessary to obtain alcohols using thermophilic bacteria, if there are industrial methods for producing alcohols using yeast. In my opinion, it is necessary to analyze the information about the strain 39ET that has been known since the end of the last century, when it was widely studied as a potential biotechnological agent for ethanol production.

We have added some text regarding the utility of thermophiles for bioprocessing an added more historical information regarding 39E’s extensive history in biotechnology.

References are given very carelessly in the text. So, in the sentence (Lines 53-54) “Thermophilic anaerobic bacteria within the genera of Thermoanaerobacter, Thermoanaerobacterium, and Caldanaerobacter have been known for some time for being good ethanol and hydrogen producers” [11–13] no mention of Geobacillus thermoglucosidasius from ref 13.

  1. Cripps, R.E.; Eley, K.; Leak, D.J.; Rudd, B.; Taylor, M.; Todd, M.; Boakes, S.; Martin, S.; Atkinson, T. 84 Metabolic engineering of Geobacillus thermoglucosidasius for high yield ethanol production. Eng. 85 2009, 11, 398–408, doi:10.1016/j.ymben.2009.08.005.

  1. thermoglucosidaius is now explicitly mentioned in text.

Line 117 Growth medium is glucose and peptone free and is unlikely to support the growth of such saccharolytic bacterium as T. pseudoethanolicus strain 39ET.

The medium does not contain peptone but does contain glucose which is now more clearly stated. The inclusion of peptone has been shown to result in the appearance of other carboxylic acids associated with the fermentation of BCAAs which can, in the presence of sufficient reducing potential, be converted to their corresponding alcohols.

Line 125-126 T. pseudoethanolicus strain 39ET = DSM 2355T type strain of species.

This is now clarified.

Line 146 Effect of initial pH on carboxylic acid reduction - how the initial pH was set? How did the pH change at the end of cultivation?

pH was adjusted using 6 M HCl or NaOH prior to autoclaving and verified after autoclaving. Given the high buffer capacity of the medium (equivalent to 50 mM phosphate), the pH did not change more than 0.3 pH units at the end of fermentation.

Line 131 What medium was used?

We now specify basal mineral (BM) medium

Line 136 What medium was used for kinetic experiments?

All cultivations were performed in BM medium as clarified in Line 131

line 142 and 147. What is BM?

This is now clarified in Line 131

Why did the cultivation continue for 5 days if the doubling time of the cell population of strain 39E is 75 min at 65oC? On all curves after 40 hours the optical density dropped, which indicates sporulation or cell lysis.

The rational for continuing the kinetic experiment for five days was to ensure that equilibrium was obtained for all processes of interest (out experience with amino acid catabolism has shown that 5 days is not always enough with the reduction of the carboxylic acids from BCAA metabolism often taking in excess of 10 days to reach their maximum concentration). Obviously there is no practical reason to cultivate this strain for this long under these conditions as 48 hours would appear to be more than sufficient based on the data. Additionally, many of the other experiments in this study were conduced for 5 days to ensure that the cultures were outgrown.

Cultivation with yeast extract as a carbon source cannot be used as control because yeast extract does not support the growth of the strain.

Yeast extract does support the growth of this strain as can be seen in Supplemental Figure 1 with a small increase in optical density and acetate formation. BM with only YE was used as a control to quantify any carboxylic acids produced from BCAA metabolism although these values were below the 0.5 mM.

Line 207 It is known that lactate is another product of glucose fermentation by strain 39ET. Was lactate formed in your experiments?

We have often observed traces of lactate when cultivating 39E although it was not analyzed in this case. Carbon balances for the vast majority of experiment was in excess of 90%.

As known strain 39ET ferments not only glucose, but carbohydrates include xylose, cellobiose, starch, maltose and sucrose. What is known about the formation of alcohols from the corresponding organic acids in the presence of other carbohydrates?

 We have done some work on this strain using other sources of reducing potential; unsurprisingly, T. pseudoethanolicus can utilize a wide range of carbon sources as source of reducing potential for carboxylic acid reduction although glucose thus far seems to be gives the best result.

From page 15, there is no line numbering and references are not drawn up according to the rules.

The line numbering issue is due to a continuous break on page 14… this has been fixed

In the references, about 23% of references related to the studies of the authors; I don’t think this is acceptable.

We´ve carefully reviewed our use of self references; they do not seem to be inappropriate although that might be a call for the editor. With the added references in the introduction, the percentage of self references is now lower.

In my opinion, the text should be rewritten vastly. Only in this case can it be accepted.

Round 2

Reviewer 2 Report

The authors' arguments about the strain cultivation for five days do not look very convincing. But the necessary clarifications have been made to the text and it can be submitted.